# Experimental identification of aminomethanol (NH$_2$CH$_2$OH)—the key intermediate in the Strecker Synthesis

Santosh K. Singh[1,2], Cheng Zhu[1,2], Jesse La Jeunesse[1,2], Ryan C. Fortenberry [3✉] & Ralf I. Kaiser [1,2✉]

The Strecker Synthesis of (a)chiral α-amino acids from simple organic compounds, such as ammonia (NH$_3$), aldehydes (RCHO), and hydrogen cyanide (HCN) has been recognized as a viable route to amino acids on primordial earth. However, preparation and isolation of the simplest hemiaminal intermediate – the aminomethanol (NH$_2$CH$_2$OH)– formed in the Strecker Synthesis to even the simplest amino acid glycine (H$_2$NCH$_2$COOH) has been elusive. Here, we report the identification of aminomethanol prepared in low-temperature methylamine (CH$_3$NH$_2$) – oxygen (O$_2$) ices upon exposure to energetic electrons. Isomer-selective photoionization time-of-flight mass spectrometry (PI-ReTOF-MS) facilitated the gas phase detection of aminomethanol during the temperature program desorption (TPD) phase of the reaction products. The preparation and observation of the key transient aminomethanol changes our perception of the synthetic pathways to amino acids and the unexpected kinetic stability in extreme environments.

[1] Department of Chemistry, University of Hawaii, 2545 McCarthy Mall, Honolulu, HI 96822, USA. [2] W. M. Keck Research Laboratory in Astrochemistry, University of Hawaii, 2545 McCarthy Mall, Honolulu, HI 96822, USA. [3] Department of Chemistry and Biochemistry, University of Mississippi, 322 Coulter Hall, University, MS 38677-1848, USA. ✉email: r410@olemiss.edu; ralfk@hawaii.edu

Since 1850, the Strecker Synthesis—a series of chemical reactions synthesizing (a)chiral α-amino acids from ammonia ($NH_3$), aldehydes (RCHO), and hydrogen cyanide (HCN) (Fig. 1)[1–5] has received extensive recognition as a conceivable route to amino acids such as glycine on primordial Earth[6–14]. Even so, the preparation and identification of one of the key intermediate of the Strecker Synthesis to the simplest amino acid glycine ($H_2NCH_2COOH$)—the aminomethanol ($NH_2CH_2OH$) hemiaminal—has been elusive[15–17]. Earlier attempts to synthesize aminomethanol ($NH_2CH_2OH$; **1**; Fig. 2) in laboratories were unsuccessful due to the facile decomposition to water ($H_2O$) and methanimine ($CH_2NH$), which eventually converts to hexamethylenetetramine ($C_6H_{12}N_4$), in aqueous media[15]. However, computational investigations revealed that **1** is kinetically stable toward dehydration to methanimine in the gas phase with a substantial barrier between 230 and 234 kJ mol$^{-1}$ separating aminomethanol from the unimolecular decomposition products[17,18]. The gas phase reactions of water ($H_2O$) with methanimine ($CH_2NH$) and ammonia ($NH_3$) with formaldehyde ($H_2CO$) to yield **1** involve high energy barriers of 194–252 kJ mol$^{-1}$ and 130–171 kJ mol$^{-1}$, respectively (reactions (5) and (1); Fig. 1)[17,18]. The sublimation of reaction products formed through a thermal chemistry of ammonia ($NH_3$) and formaldehyde ($H_2CO$) in water ($H_2O$) rich ices resulted in ion counts at mass-to-charge ($m/z$) of 47, but the ionization of the neutrals by electron impact could not assign the nature of the structural isomer(s)[19]. An alternative pathway to **1** could comprise the barrierless insertion of an electronically excited oxygen atom (O($^1D$)) into a carbon-hydrogen bond of the methyl group of methylamine (reaction (6); Fig. 1)[20–22] followed by stabilization of the aminomethanol insertion product either by a third body collision in the gas phase or through energy transfer processes to the surrounding molecules within an icy matrix. Furthermore, the rate of unimolecular decomposition of aminomethanol ($NH_2CH_2OH$) to methanimine ($CH_2NH$) will significantly reduce in a non-aqueous medium containing reactants methylamine and oxygen. In a water-rich environment, the surrounding water molecules could act as a catalyst in the dehydration process of aminomethanol[23] and therefore could obscure its detection.

In this work, we report the preparation of the previously elusive transient aminomethanol ($NH_2CH_2OH$; **1**) in low-temperature binary ices of methylamine ($CH_3NH_2$) and oxygen ($O_2$) along with its gas phase detection. Isomer-selective photoionization reflectron time-of-flight mass spectrometry and isotopic substitution experiments confirmed the synthesis of **1** in our molecular system. Barrierless insertion of O($^1D$) into a carbon-hydrogen bond of the methyl group of the methylamine molecule leads to the formation of **1** followed by stabilization in the ice matrix. The methodology presented here could also be useful to prepare higher order hemiaminals (RCH(OH)NH$_2$) from corresponding primary or secondary amines.

## Results

**Experimental scheme**. The experiments were performed in an ultrahigh vacuum (UHV) surface science machine at a base pressure of $9 \pm 1 \times 10^{-11}$ Torr (Supplementary Fig. 1)[24,25]. Binary ices of methylamine ($CH_3NH_2$)–oxygen ($O_2$), d5-methylamine ($CD_3ND_2$)–oxygen ($O_2$), and methylamine ($CH_3NH_2$)– O18-oxygen ($^{18}O_2$) were prepared in separate experiments at thicknesses of $239 \pm 24$ nm at $5.0 \pm 0.2$ K and oxygen to methylamine ratios of $9 \pm 1$: 1 (Supplementary Methods). Bond-cleavage processes in the ice mixtures were triggered by exposing the oxygen-rich ices to energetic electrons at average doses of typically $18 \pm 2$ eV molecule$^{-1}$ (Supplementary Table 1). Fourier Transform Infrared (FTIR) spectra of the ices were collected online and in situ during the radiation exposure to monitor chemical changes in the systems (Supplementary Fig. 2, Supplementary Table 2). The stretching frequencies of hydroxy (−OH) and amino (−NH$_2$) functional groups of aminomethanol are expected to appear at 3659 and 3146 cm$^{-1}$ (scaled), respectively, (Supplementary Fig. 2). However, broad absorption in the region 3680–3000 cm$^{-1}$ due to overlapping infrared absorptions of products carrying amino (–NH$_2$) and hydroxy functional groups (–OH) obscure the explicit identification of aminomethanol via FTIR spectroscopy (Supplementary Table 2) in these multi-component ice mixtures. Therefore, a novel methodology is required to identify isomers of unstable molecules in complex molecular mixtures selectively. Here, the products formed in the irradiated ices along with the reactants were sublimed by increasing the temperature of the sample to 320 K at a rate of 1 K min$^{-1}$ (temperature program desorption; TPD). The subliming neutral molecules were ionized exploiting isomer-selective vacuum ultraviolet (VUV) photoionization (PI) (Supplementary Table 3)[26–30]. The ions formed were then mass resolved in a reflectron time-of-flight mass spectrometer (PI-ReTOF-MS). By systematically tuning the photon energies below and above the ionization energy (IE) of the isomer of interest, isomers (**1–5**) of aminomethanol can be selectively ionized based on their adiabatic ionization energies to ultimately elucidate which isomer(s) is (are) formed. The computed molecular structures of distinct $CH_5NO$ isomers **1–5** are depicted in Fig. 2 along with their adiabatic ionization energies (IEs), relative energies, and geometrical parameters. The IEs and relative energies of the isomers calculated at the CCSD(T)/CBS level of theory with CCSD(T)/aug-cc-pVTZ zero-point vibrational energy corrections are provided in Supplementary Table 4. Optimized geometrical coordinates and calculated frequencies of the isomers are provided in Supplementary Data 1 and Supplementary Data 2, respectively. It is important to stress here that the calculated IEs of the isomers can be lower by 0.05 eV or higher by 0.02 eV based on a comparison of experimental and calculated IEs of molecular benchmark systems[31]. Further, ReTOF-MS calibration experiments revealed that the electric field of the ion optics lowers the IE by 0.03 eV[32].

**Mass spectrometry**. The mass spectra of the species subliming from the irradiated ice mixtures are collected as a function of temperature at photon energies of 9.50 and 9.10 eV (Fig. 3). At 9.50 eV, all isomers (**1**)–(**5**)—if formed—can be ionized; at

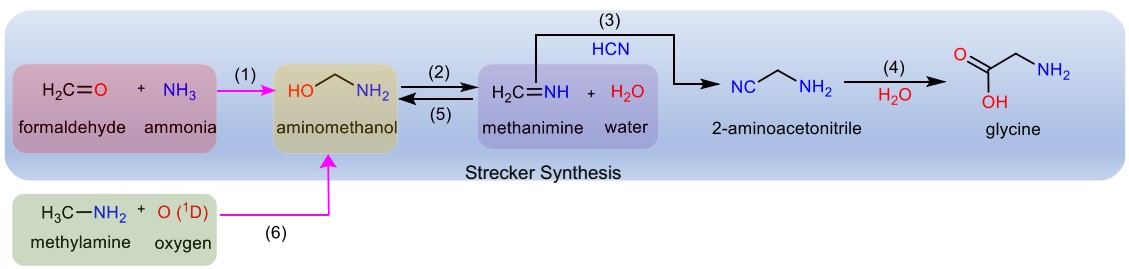

**Fig. 1 Strecker synthesis of glycine.** A simplified view of the reaction steps (1–4) involved in the Strecker Synthesis of glycine. Pink arrows indicate the proposed laboratory synthetic route (1, 6) to aminomethanol.

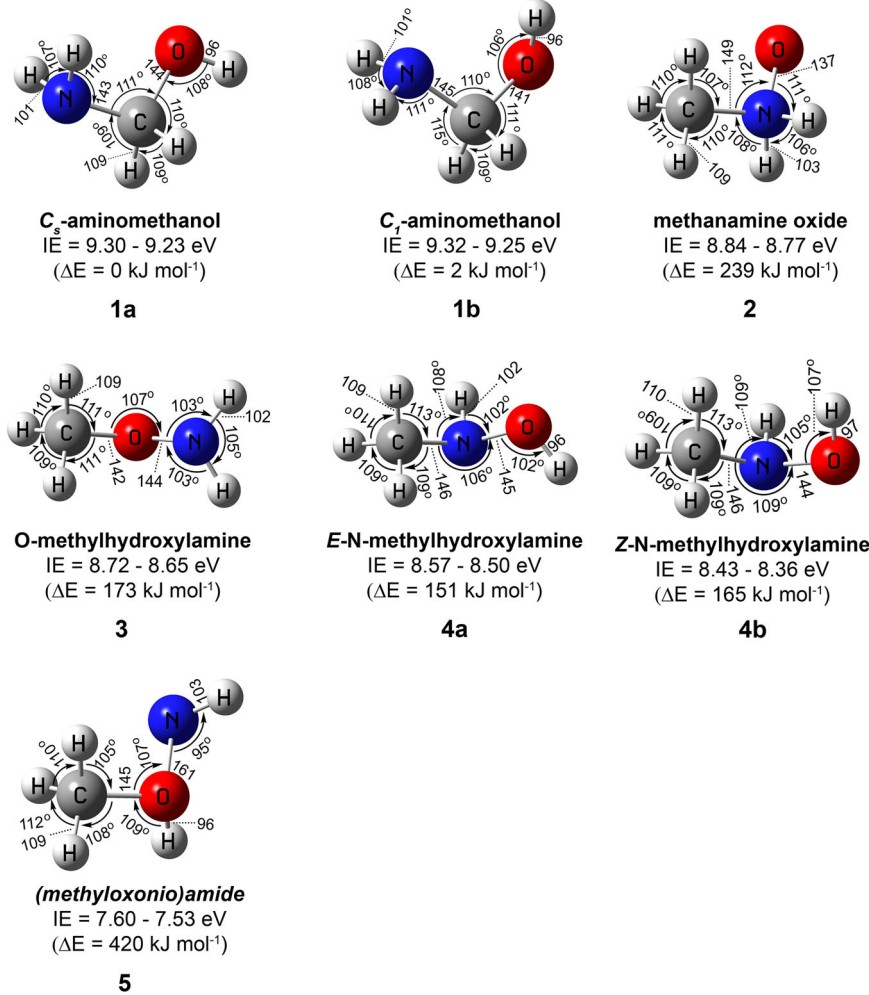

**Fig. 2 Structures of CH₅NO isomers.** Ranges of calculated adiabatic ionization energies (IEs) accounting for the error limits and relative energies (ΔE) calculated at the CCSD(T)/CBS//CCSD(T)/aug-cc-pVTZ level of theory in eV and kJ mol⁻¹, respectively. Geometrical parameters, bond length, and angles are provided in picometer and degrees, respectively. Cartesian coordinates of the structures and their calculated vibrational frequencies are provided in Supplementary Data 1 and 2, respectively.

9.10 eV, all isomers *except* (**1**) can be ionized. Data recorded at 9.50 eV reveal prominent product signal at mass-to-charge ratios ($m/z$) of 42 (CH₂N₂), 43 (C₂H₅⁺), 45 (CH₃NO/C₂H₇N), 47 (CH₅NO), 58 (C₂H₆N₂), 59 (C₃H₉N), 60 (C₂H₈N₂), 61 (CH₃NO₂), 74 (C₂H₆N₂O), and 90 (C₂H₆N₂O₂/C₃N₃H₁₂) along with the reactant signal at m/z = 31 (CH₃NH₂). Signals at $m/z = 60$ (C₂H₈N₂), 59 (C₃H₉N), and 58 (C₂H₆N₂) were previously observed in processed pure methylamine ice and can be tentatively assigned to isomers of ethylene diamine, *N*-methyl-ethanamine and *N*-methylformimidamide[33]. A possible formation pathway of ethylene diamine (C₂H₈N₂) could follow a radical–radical recombination of two •CH₂NH₂ radicals[33]. *N*-methyl-ethanamine (C₃H₉N) could generate via recombination of •CH₃ and •CH₂NHCH₃ radicals, while *N*-methylformimidamide (C₂H₆N₂) could generate via dehydrogenation of ethylene diamine (C₂H₈N₂). Oxygen could insert at the C–H and/or N–H bonds of N-methylformimidamide to generate products observed at $m/z = 74$ (C₂H₆N₂O) and 90 (C₂H₆N₂O₂). It is important to note here that mass signals 47, 61, 74, and 90 were not observed in pure methylamine ices[33].

In a control experiment, in which PI-ReTOF-MS data were collected under the same experimental conditions, but without exposing the ices to energetic electrons, no product signal was observed (Supplementary Fig. 3). Therefore, in the aforementioned experiments, signal detected at $m/z = 45$, 47, 58, and 60 can be only linked to the processing of the ices by energetic electrons, but not to ion-molecule reactions in the gas phase of the reactant molecules. The TPD profile at $m/z = 47$ recorded at a photon energy of 9.50 eV reveals a single sublimation event peaking at 210 K (Fig. 4a). This signal can be associated with a product of a molecular formula CH₅NO. Recall that at 9.50 eV, all the isomers **1–5** can be photoionized. Upon lowering the photon energy to 9.10 eV—a photon energy where only isomers **2–5** can be ionized—no ion counts could be observed (Fig. 4b). These data document that the ion signal observed at $m/z = 47$ is connected to the previously elusive aminomethanol conformers **1a-b**; the latter could not be discriminated since their ionization energies overlap within the error limits. To confirm the molecular formula of the product, isotopic substitution experiments were conducted with CH₃NH₂–¹⁸O₂ and CD₃ND₂–O₂ ice mixtures. The peak sublimation temperature of the TPD profiles collected at $m/z = 49$ and 52 in the CH₃NH₂–¹⁸O₂ and CD₃ND₂–O₂ systems, respectively, are in excellent agreement with the peak sublimation temperature of 210 K recorded for $m/z = 47$ in the CH₃NH₂–O₂ system (Fig. 5); this also confirms a shift of the ion signal ($m/z = 47$) by 2 amu and 5 amu revealing the presence of a single oxygen atom and five hydrogen atoms. The broadening of the TPD profile at $m/z = 49$ in the CH₃NH₂–¹⁸O₂ ice close to 200 and 230 K could be due to

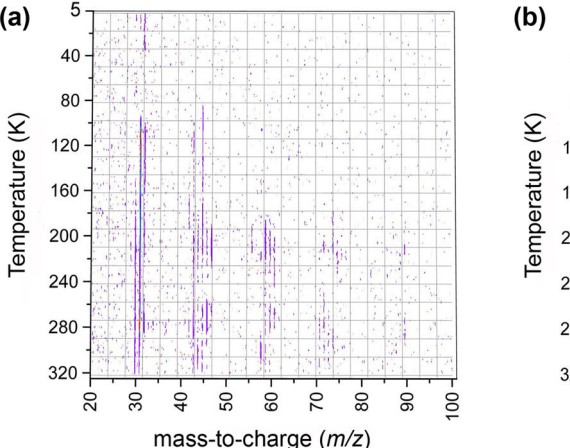
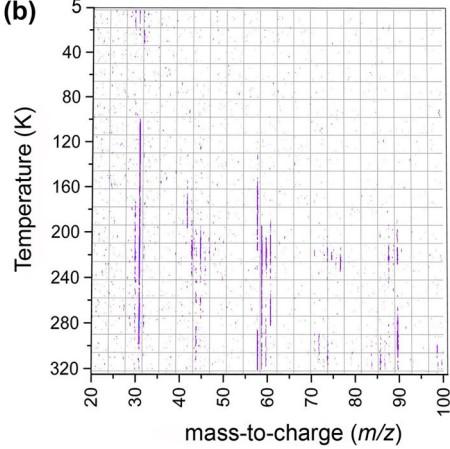

**Fig. 3 PI-ReTOF mass spectra.** Measured as a function of temperature during the temperature program desorption phase of irradiated methylamine (CH$_3$NH$_2$)-oxygen (O$_2$) ice mixtures at photon energies of (**a**) 9.50 and (**b**) 9.10 eV. The dark purple colored lines indicate ion counts. Source data are provided as a Source Data file.

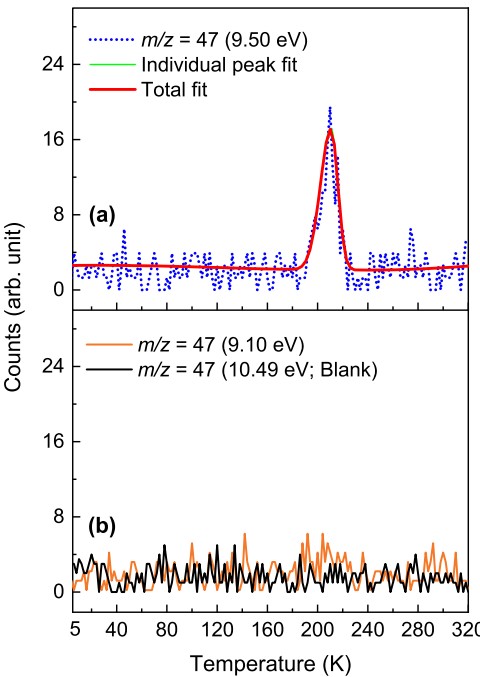

**Fig. 4 PI-ReTOF mass spectra collected at *m/z* = 47 as a function of temperature. a** Measured at 9.50 eV photon energy during the sublimation phase of irradiated methylamine-oxygen ices. **b** Recorded at 9.10 and 10.49 eV photon energies during the sublimation phase of irradiated and non-irradiated ice mixtures, respectively. Source data are provided as a Source Data file.

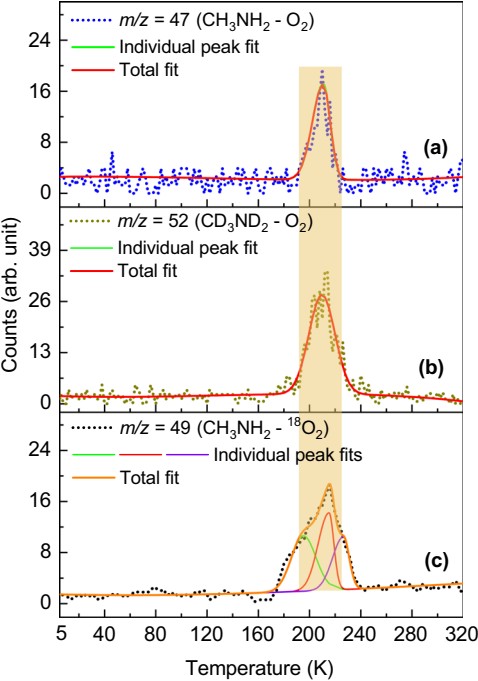

**Fig. 5 PI-ReTOF mass spectra collected in isotopic experiments at a photon energy of 9.50 eV. (a)**, (**b**), and (**c**) are temperature program desorption (TPD) profiles obtained at *m/z* = 47, 52, and 49 during the sublimation phase of irradiated CH$_3$NH$_2$-O$_2$, CD$_3$ND$_2$-O$_2$, and CH$_3$NH$_2$-$^{18}$O$_2$ ice mixtures, respectively. The shaded region indicates the peak corresponding to the target molecule, aminomethanol. Source data are provided as a Source Data file.

the fragmentation of higher masses observed at *m/z* = 91 (C$_2$H$_3$N$_2$$^{18}$O$_2$) and 80 (CH$_4$N$_2$$^{18}$O$_2$) (Supplementary Fig. 4). Product observed at mass 91 could fragment to CH$_2$N$_2$$^+$ (*m/z* = 42) and HCO$_2$$^+$ (*m/z* = 49) ions, the later fragment could add to the ion counts at *m/z* = 49 around 200 K. Similarly, product observed at *m/z* = 80 could fragment to generate N$_2$H$_3$$^+$ (*m/z* = 31) and HCO$_2$$^+$ (*m/z* = 49) ions; the later fragment will increase the ion counts at *m/z* = 49 near 230 K causing broadening of the TPD profile of our target molecule. In CH$_3$NH$_2$–O$_2$ and CD$_3$ND$_2$–O$_2$ ice systems, the ion fragment HCO$_2$$^+$ will

appear at *m/z* ratio of 45 and 46, respectively, and therefore cannot appear in the mass channel of our target molecule.

## Discussion
Having provided compelling evidence on the synthesis and detection of aminomethanol in the gas phase, we shift our attention now to the identification of potential reaction mechanism(s) involved in its formation. Previous investigations of neat oxygen (O$_2$) ices exposed to energetic electrons revealed

two decomposition pathways leading to two ground-state atomic oxygens [O ($^3P$)] and one ground state O ($^3P$) plus an electronically excited O($^1D$) atom (reactions (7) and (8))[34]. O($^1D$) can insert barrierlessly into a carbon-hydrogen bond of the methyl group of the methylamine molecule forming aminomethanol followed by stabilization in the icy matrix (reaction (9))[20,21]. Alternatively, methylamine (CH$_3$NH$_2$) can undergo unimolecular dissociation to form the aminomethyl radical (•CH$_2$NH$_2$) radical and atomic hydrogen upon exposure to ionizing radiation (reaction (10))[33,35], which may recombine with atomic oxygen to the hydroxyl radical (OH) (reaction (11)). The latter could undergo radical–radical recombination with •CH$_2$NH$_2$ radical if they hold a favorable recombination geometry to form aminomethanol (reaction (12)). The present experiments cannot discriminate between the insertion versus radical–radical recombination pathway.

$$O_2 \rightarrow O(^3P) + O(^3P) \qquad \Delta_R G = 494 \, kJ \, mol^{-1} \qquad (7)$$

$$O_2 \rightarrow O(^3P) + O(^1D) \qquad \Delta_R G = 683 \, kJ \, mol^{-1} \qquad (8)$$

$$CH_3NH_2 + O(^1D) \rightarrow NH_2CH_2OH \quad \Delta_R G = -617 \, kJ \, mol^{-1} \quad (9)$$

$$CH_3NH_2 \rightarrow CH_2NH_2 + H \qquad \Delta_R G = 382 \, kJ \, mol^{-1} \qquad (10)$$

$$O(^3P) + H \rightarrow OH \qquad \Delta_R G = -426 \, kJ \, mol^{-1} \qquad (11)$$

$$CH_2NH_2 + OH \rightarrow NH_2CH_2OH \quad \Delta_R G = -383 \, kJ \, mol^{-1} \quad (12)$$

In conclusion, we prepared aminomethanol **1** in methylamine-oxygen ices upon exposure to ionizing radiation and detected this novel molecule during the sublimation phase utilizing isomer-selective photoionization reflectron time-of-flight mass spectrometry (PI-ReTOF-MS) along with isotopic substitution experiments. Considering the distance between the ice surface and the photoionization laser of $2.0 \pm 0.5$ mm along with the average velocity of 307 m s$^{-1}$ at 210 K, the lifetime of **1** in the gas phase has to exceed $6.5 \pm 1.5$ μs to survive the flight time between the ice surface and photo ionizing region. The kinetic stability of **1** is further reflected by the high decomposition energy barriers of 169 and 235 kJ mol$^{-1}$ to formaldehyde (H$_2$CO) and ammonia (NH$_3$) and methanimine (CH$_2$NH) and water (H$_2$O), respectively (Supplementary Fig. 5, Supplementary Table 5). The approach presented here could be helpful to prepare higher order transient hemiaminals like RCH(OH)NH$_2$ with 'R' being an organic group via O($^1D$) insertion into the carbon-hydrogen bond of the CH$_2$ moiety of the corresponding amines (RCH$_2$NH$_2$).

The present study also shows evidence of the formation of aminomethanol in astrophysical-like conditions. The methylamine (CH$_3$NH$_2$)-oxygen (O$_2$) ice mixture can be appraised as a model astrophysical ice-analog. Primary amines such as methylamine (CH$_3$NH$_2$) and ethylamine (C$_2$H$_5$NH$_2$) were observed in interstellar medium (ISM) towards SgrB2 as well as on comet 67 p/Churyumov-Gerasimenko[36–38]. Amines are also contemplated to play a critical role in the prebiotic chemistry of early Earth[39,40]. Analysis of the samples returned from the *stardust* mission revealed the presence of methylamine, ethylamine, along with glycine[41]. Although methylamine (CH$_3$NH$_2$) has only been detected in the gas phase of the interstellar medium, laboratory experiments revealed that CH$_3$NH$_2$ could be formed via recombination of methyl (•CH$_3$) and amino (•NH$_2$) radicals in a matrix cage[42]. Methane (CH$_4$) and ammonia (NH$_3$), which are precursors of •CH$_3$ and •NH$_2$ radicals, have been detected on icy grains;[43] hence methylamine (CH$_3$NH$_2$) is expected to exist on interstellar grains, albeit at a lower concentration[44]. Oxygen (O$_2$) was detected in the ISM much later due to observational difficulties and probably low abundance in the gas phase of ISM[45]. It

has been suggested that the poor abundance of oxygen in the gas phase could be due to the condensation of molecular oxygen (O$_2$) onto the interstellar grains[46]. Although the exact composition of molecular oxygen in the ice grains has not yet been determined, the possibility of O$_2$ being an active reactant in the ISM cannot be neglected. Very recently, Bergner et al. investigated the formation of methanol in methane (CH$_4$) and oxygen (O$_2$) ice mixture and reported that oxygen insertion chemistry should be efficient on grain surfaces in cold interstellar regions such as proto-stellar cores and protoplanetary disk midplanes[47].

Our laboratory experiments in astrophysical-like conditions revealed that aminomethanol could generate from amines on ice grains in proto-stellar cores or protoplanetary disks and subsequently incorporated inside comets or (and) meteorites[48], which is (are) source(s) of extraterrestrial amino acids on early Earth. Aminomethanol (NH$_2$CH$_2$OH) could produce amino acid precursor—aminoacetonitrile (NH$_2$CH$_2$CN) in the presence of hydrogen cyanide; the later molecule could eventually produce amino acids on comets and meteorites through the Strecker route upon aqueous alteration. It is important to note here that the Strecker process could initiate even in the absence of liquid water. Quantum chemical calculations performed by Rimola *et al.* show that a Strecker-type reaction to form glycine is energetically feasible on solid-water ice grains[23]. The formation of acetonitrile precursor of glycine on an icy grain mantle via the addition of hydrogen cyanide (HCN) to a dehydrogenated form of aminomethanol has also been shown through calculations[49]. The present experimental study supports the theoretical investigations[23,49] that have demonstrated the involvement of aminomethanol in Strecker synthesis of glycine in astrophysical environments. The current research also conclusively shows that the amines could represent textbook candidates for the Strecker synthesis of amino acids via the formation of aminoalcohols in the prebiotic soup on Earth and in deep space.

## Methods

**Experimental**. Experiments were conducted in an ultrahigh vacuum chamber evacuated to a base pressure of $9 \pm 1 \times 10^{-11}$ torr using turbo molecular pumps backed by a dry scroll pump (Supplementary Fig. 1)[24,25]. To prepare each ice mixture, gases of methylamine (CH$_3$NH$_2$; 99.95% Matheson TriGas) and oxygen molecule (O$_2$; 99.99% Sigma Aldrich) were premixed and deposited via a glass capillary array onto a silver substrate at an angle of 20° with respect to the normal of the substrate. The silver substrate is mounted on a cold finger made from oxygen-free high conductivity copper. The temperature of the cold finger was maintained at $5.0 \pm 0.2$ K using a closed cycle helium refrigerator during deposition of the gases. The thickness of each ice ($d_i$; nm) was monitored online and in situ via laser interferometry. In this method, a He-Ne laser of 632.8 nm wavelength (λ) has been used at an angle of incidence (θ) equal to 4° to measure the interference pattern formed after reflection of the laser light from the silver substrate and ice surface. One interference fringe ($m = 1$) was observed during the deposition of the ice. The average refractive index of the ice mixture ($n_{ice} = 1.32$) was determined from the refractive indices of the neat oxygen ($n = 1.25$)[50] and methylamine ices ($n = 1.40$)[35] reported in the literature. The thickness of the ice mixture was determined to be $239 \pm 24$ nm using equation (3) provided in the Supplementary Methods. The infrared spectra of the ice mixtures were collected in the 4000–700 cm$^{-1}$ region using a Fourier Transform Infrared Spectrometer (Nicolet 6700) operated at a resolution of 4 cm$^{-1}$ in an absorption-reflection-absorption mode with an incidence angle of 45° (Supplementary Fig. 2). The ratio of oxygen molecule (O$_2$) to methylamine (CH$_3$NH$_2$) in the ice mixture was found to be $9.2 \pm 1.0$: 1, based on the column densities (N; molecules cm$^{-2}$) of O$_2$ and CH$_3$NH$_2$ (Supplementary Methods). Thereafter, each ice mixture was exposed to 5 keV energetic electrons at an electron current of $100 \pm 10$ nA for 60 min. IR spectra of the ices were measured in situ during irradiation to monitor the changes induced by ionizing radiation. Astrophysical ices experience high energy radiations i.e. UV photons and cosmic-charged particles. The galactic cosmic ray field consists predominantly of protons with distribution maximum of about 10 MeV and losses 99.9 % of their kinetic energy to the electron system of the target molecules[51]. This electronic energy transfer generates energetic electrons (δ-electrons), which could penetrate into the ice up to a few hundred nanometers and initiate bond rupture processes in the molecular species of the ice mixture via inelastic energy transfer. In this experiment, we have used 5 keV electrons to mimic the effects of δ-electrons. The linear electron energy transfer of MeV protons to the ice target holds a similar value as the 5 keV electrons[51,52]. Thus, our experiments simulate the formation of

aminomethanol in the ice mixture of methylamine and oxygen via charged particles through electronic energy transfer. Using Monte Carlo simulations via CASINO 2.42 software[53], the average energy dose was calculated to be $18.3 \pm 2.0$ eV molecule$^{-1}$ for methylamine and $18.9 \pm 2.3$ eV molecule$^{-1}$ for oxygen (Supplementary Table 1). Hereafter, ices were annealed at a rate of 1 K min$^{-1}$, and molecules subliming from the substrate were ionized and detected using photoionization along with reflectron time-of-flight mass spectrometry (PI-ReTOF-MS). Pulsed VUV light is utilized for the photoionization of the molecules. Two VUV energies at 9.50 and 9.10 eV were used to identify the isomers (Supplementary Table 3). The ions formed are extracted and eventually separated based on their mass-to-charge (m/z) ratio before reaching to microchannel plate (MCP) detector. The MCP detector generates a signal when ions reach to the detector. This signal is amplified using a preamplifier (Ortec 9305) and shaped with a 100 MHz discriminator. The discriminator sends the signal to a computer based multichannel scaler, which records the signal in 4 ns bins triggered at 30 Hz by a pulse delay generator. 3600 sweeps were collected for each mass spectrum per 1 K increase in the temperature during the TPD phase.

**Computational**. We have employed the density functional theory (DFT) B3LYP/aug-cc-pVTZ geometry optimizations and subsequent harmonic zero-point vibrational energy (ZPVE) corrections from the Gaussian16 program[54] with coupled cluster singles, doubles, and perturbative triples [CCSD(T)] aug-cc-pVTZ and aug-cc-pVQZ[55,56] single-point energies extrapolated to the complete basis set (CBS) limit via a two-point formula[57] in the MOLPRO 2019.2 quantum chemistry program[58]. The higher-level of theory (utilized for comparison) exclusively employs CCSD(T) and MOLPRO where the geometries are optimized and harmonic frequencies (along with ZPVEs) computed with the aug-cc-pVTZ basis set. At these geometries, CCSD(T)/aug-cc-pVQZ single-point energies are computed and the same two-point extrapolation scheme is employed to provide the adiabatic ionization energies and relative energies. All radical cations make use of restricted open-shell Hartree Fock reference wavefunctions, and all neutrals employ fully restricted Hartree Fock references. The transition state optimized geometries and harmonic frequencies are computed with MP2 theory and the aug-cc-pVTZ basis set within Gaussian16 where subsequent CCSD(T)/aug-cc-pVTZ and aug-cc-pVQZ single-point energies are extrapolated to the CBS limit in the same fashion as that done in the other cases.

## Data availability

The data generated in this study are provided in the main Article, Supplementary Information, and Supplementary Data files. Source data are provided with this paper.

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

## Acknowledgements
This work was supported by the US National Science Foundation, Division of Astronomical Sciences under grants AST-1800975 and AST-2103269 awarded to the University of Hawaii. R.C.F. acknowledges support from NASA grant NNX17AH15G, startup funds provided by the University of Mississippi, and computational support from the Mississippi Center for Supercomputing. Research funded in part by NSF Grants CHE-CHE-1338056 and OIA-1757220.

## Author contributions
R.I.K. and S.K.S. designed the experiments. S.K.S., C.Z. and J.L.J. performed the experiments. R.C.F. performed the calculations. S.K.S. and R.I.K. wrote the manuscript, which was revised and approved by all co-authors.

## Competing interests
The authors declare no competing interests.
