## [Peer Review File · Nature Communications]

REVIEWER COMMENTS

Reviewer #1 (Remarks to the Author):

The authors present their original discovery of aminomethanol synthesized during the electron irradiation of methylamine – oxygen ices. Unambiguous detection of aminomethanol is achieved by a unique combination of two standard techniques: temperature program desorption and photoionization time of flight mass spectrometry. Comparison of the sublimation temperature dependence of the mass spectra for irradiated ices containing either the isotopologue CD₃ND₂ – O₂, or CH₃NH₂ - ¹⁸O₂, or aminomethanol – oxygen ice demonstrates the unambiguous detection of aminomethanol that is clearly produced by electron-molecule interactions in the simple ice mixture.

Using standard quantum chemistry tools, the authors briefly discuss two reaction mechanisms, either of which may be responsible for the production of aminomethanol: the electron-impact dissociation of O₂, producing an excited oxygen atom that reacts with methylamine to form aminomethanol in the icy matrix; or the unimolecular dissociation of methylamine in the ice, releasing a hydrogen atom, that combines with an oxygen atom before recombining with the aminomethyl radical. Both mechanisms seem feasible, and perhaps future electron scattering calculations, as well as molecular dynamics simulations will shed more light on the relative likelihood of each mechanism.

This work is of a broad interest to the scientific community due to the fundamental importance of this synthetic pathway in the abiotic/prebiotic synthesis of simple amino acids. The paper is very clearly written, and the conclusions are fully-supported by the experimental results. I recommend publication in Nature Communications without further revision.

Reviewer #2 (Remarks to the Author):

This study considers the formation of aminomethanol in mixed O₂/NH₂CH₃ laboratory ices, through electron bombardment. This process is intended to reproduce the effects of cosmic ray impingement on interstellar ices and/or solar system bodies. The electron-induced dissociation of O₂ results in the production of singlet-D oxygen, which has been shown elsewhere to be capable of insertion into C-H bonds; insertion into NH₂CH₃ is found here to produce aminomethanol (NH₂CH₂OH). Aminomethanol is an intermediary in the Strecker synthesis of glycine and/or other amino acids, and its possible presence in interstellar space could therefore enable the production of amino acids

either in situ, or following delivery to planetary surfaces. But its production in the laboratory has proved to be very challenging. The present work cleverly uses photoionization mass spectrometry during temperature programmed desorption of the post irradiation ices to detect the presence of aminomethanol in the gas phase. The work makes use of calculated ionization potentials to distinguish between the different CH₅NO isomers.

I believe that this is an important and interesting enough study to be published in Nature Communications. In fact, I believe it somewhat undersells the ingenuity of the successful technique presented here, providing as it does the evidence of the laboratory production of this elusive molecule. A little more could be added to the text to put the implications of the work in context for the non-expert reader (see below), as well as to explain more clearly how important the result may be for amino acid production in pre-solar or early solar system environments. It is my feeling that the same experiment conducted using an ice mixture more representative of interstellar ices might not produce enormous quantities of aminomethanol, so the influence of the Strecker mechanism in feeding off of aminomethanol formed in precisely this way might not be so great (amino acetonitrile could be formed through other mechanisms, for example). The Strecker mechanism should also arguably take place in the liquid phase, at least during the hydrolysis stage, so interstellar production in this way may not be so plausible. This should be commented on. Cometary, meteoritic or early Earth conditions may nevertheless lend themselves to amino acid production in this way.

Besides the above, I list a selection of comments, questions and suggestions:

1/ Considering who may be reading this work, I suggest that the title indicate clearly that the identification of aminomethanol has taken place in an experimental setting (rather than in space).

2/ The relevance of electron bombardment in the interstellar context is not addressed. It should be briefly explained how these electrons are understood to come about and what is known about their energy spectrum.

3/ What is the deposition angle of the ice? This is presumably not background deposition. The deposition angle can affect ice porosity and it is therefore relevant to state it (in exp. methods).

4/ Figures 2 and S3 are not self-explanatory. More information should be provided in the caption to enable the reader to understand what is being shown. In Fig. 2, for instance, the plotted colors extend beyond the bounds of the plot, indicating some sort of three dimensional feature. It is not stated what this dimension and the coloring actually mean.

5/ While it is experimentally interesting and laudable to have formed and identified aminomethanol, the meaning in an interstellar context is less clear, and more should be made of this. For example, interstellar ices should have substantial water content, while alternative C-H bonds for insertion by O singlet-D should also be prevalent in methane and methanol, and arguably others. We also do not know what the O₂ composition of interstellar ices actually should be, based on observations (although this is not the case for cometary ice). It could, in fact, be small; smaller even than that of NH₂CH₃, which has also not yet been detected in the solid phase in the ISM and has only recently been detected in the gas phase in rather small quantities. How might all of this impact the importance of the findings?

6/ How applicable is Strecker synthesis in a purely solid-phase context?

7/ How does the mechanism tested here compare with other lab studies, e.g. that of Elsila et al. (2007, ApJ, 600, 911), who used H₂O:CH₃OH:HCN:NH₃ mixtures to produce amino acids? Although they considered a slightly different mechanism, their isotopic labeling results indicated that Strecker synthesis should be less important than radical reaction processes.

8/ The identification of aminomethanol rests strongly on the calculations, which mainly take a back seat in the text. Are there any alternative estimates or measurements of any of the ionization energies calculated here? Is there any alternative verification that can now or in future be used to back up the calculations?

If all of the above points are addressed, I feel that the manuscript would be ready for publication.

Reviewer #3 (Remarks to the Author):

The present communication entitled "Gas-phase identification of aminomethanol (NH₂CH₂OH) – the key intermediate in the Strecker synthesis", S.K. Singh et al shows an experimental work on the formation of NH₂CH₂OH induced by keV electrons of NH₂CH₃:O₂ ice. Coupling VUV photoionization time of flight spectrometer and theoretical calculations (IE of different isomers), they have identified NH₂CH₂OH isomers desorbing from irradiated ice. This work relies on well establish procedure in Kaiser's laboratory to investigate formation of different molecules in irradiated ices.

A weak aspect of the paper is that the investigation of $\text{NH}_2\text{CH}_2\text{OH}$ formation is not a novelty from a conceptual point of view. In the literature, one can find several studies dedicated to investigate the formation of aminomethanol experimentally and theoretically (several references are in the present article). For example it has been shown that $\text{NH}_2\text{CH}_2\text{OH}$ can be formed in $\text{H}_2\text{O}:\text{NH}_3:\text{H}_2\text{CO}$ ice (F Duvernay et al. *ApJ* 791 (2014) 75). It would be worth mentioning why in particular $\text{NH}_2\text{CH}_3:\text{O}_2$ is particularly interesting to investigate aminomethanol formation.

Another point, the whole procedure of isomer identification relies on theoretical calculations. Authors considered error limits due to theoretical uncertainties and experimental conditions for benchmark systems. It would be worth also to compare with experimental IE values of discussed isomers (if available).

One can find also inconsistency between the text in "Mass spectrometry" paragraph and Fig 2. It has been stated the data recorded at 9.50 eV reveal several products at m/z 45, 47, 58 and 60. This is visible in Fig. 2b (obtained for 9.10 eV if we follow a caption)? Figure 2 is nice to follow globally dynamics of appearing and disappearing m/z species as a function of temperature. However, the present figure does not allow to read masses in details. Lines to guide eyes should be added or it would be better to show a cut for 210 K –intensity as a function of m/z for all three ices $\text{NH}_2\text{CH}_3:\text{O}_2$, $\text{Nd}_2\text{CD}_3:\text{O}_2$ and $\text{NH}_2\text{CH}_3:18\text{O}_2$. A short comment of high mass formation would enrich the article.

It would be also worth to clarify broadening of the TPD profile at $m/z = 49$. It has been mentioned that is due to fragmentation of masses $m/z = 58$ ($\text{C}_2\text{H}_6\text{N}_2$) and 80 (CH_418O_2) (e.g. loss of fragments of m/z 9 and 31?, respectively). This fragmentation is caused by which factor? Moreover, why do we not observe similar broadening for other ices?

Reviewer #1 (Remarks to the Author):

The authors present their original discovery of aminomethanol synthesized during the electron irradiation of methylamine – oxygen ices. Unambiguous detection of aminomethanol is achieved by a unique combination of two standard techniques: temperature program desorption and photoionization time of flight mass spectrometry. Comparison of the sublimation temperature dependence of the mass spectra for irradiated ices containing either the isotopologue CD₃ND₂ – O₂, or CH₃NH₂ - ¹⁸O₂, or aminomethanol – oxygen ice demonstrates the unambiguous detection of aminomethanol that is clearly produced by electron-molecule interactions in the simple ice mixture.

Using standard quantum chemistry tools, the authors briefly discuss two reaction mechanisms, either of which may be responsible for the production of aminomethanol: the electron-impact dissociation of O₂, producing an excited oxygen atom that reacts with methylamine to form aminomethanol in the icy matrix; or the unimolecular dissociation of methylamine in the ice, releasing a hydrogen atom, that combines with an oxygen atom before recombining with the aminomethyl radical. Both mechanisms seem feasible, and perhaps future electron scattering calculations, as well as molecular dynamics simulations will shed more light on the relative likelihood of each mechanism.

This work is of a broad interest to the scientific community due to the fundamental importance of this synthetic pathway in the abiotic/prebiotic synthesis of simple amino acids. The paper is very clearly written, and the conclusions are fully-supported by the experimental results. I recommend publication in Nature Communications without further revision.

Response: We thank the reviewer for his/her positive comments.

Reviewer #2 (Remarks to the Author):

This study considers the formation of aminomethanol in mixed O₂/NH₂CH₃ laboratory ices, through electron bombardment. This process is intended to reproduce the effects of cosmic ray impingement on interstellar ices and/or solar system bodies. The electron-induced dissociation of O₂ results in the production of singlet-D oxygen, which has been shown elsewhere to be capable of insertion into C-H bonds; insertion into NH₂CH₃ is found here to produce aminomethanol (NH₂CH₂OH). Aminomethanol is an intermediary in the Strecker synthesis of glycine and/or other amino acids, and its possible presence in interstellar space could therefore enable the production of amino acids either in situ, or following delivery to planetary surfaces. But its production in the laboratory has proved to be very challenging. The present work cleverly uses photoionization mass spectrometry during temperature programmed desorption of the post irradiation ices to detect the presence of aminomethanol in the gas phase. The work makes use of calculated ionization potentials to distinguish between the different CH₅NO isomers.

I believe that this is an important and interesting enough study to be published in Nature Communications. In fact, I believe it somewhat undersells the ingenuity of the successful technique presented here, providing as it does the evidence of the laboratory production of this elusive molecule. A little more could be added to the text to put the implications of the work in context for the non-expert reader (see below), as well as to explain more clearly how important the result may be for amino acid production in pre-solar or early solar system environments.

Response: We thank the reviewer for appreciating our study and the techniques used here. We have now discussed the significance of our results in the context of amino acid production in pre solar or early solar system environments in the manuscript.

Relevant changes in the manuscript (on pages 7-8): *The following paragraphs are added in the discussion section.*

“The present study also shows evidence of the formation of aminomethanol in astrophysical-like conditions. The methylamine (CH₃NH₂)-oxygen (O₂) ice mixture can be appraised as a model astrophysical ice-analog (ISM). Primary amines such as methylamine (CH₃NH₂) and ethylamine (C₂H₅NH₂) were observed in ISM towards SgrB2 as well as on comet 67 p/Churyumov-Gerasimenko.³⁶⁻³⁸ Amines are also contemplated to play a critical role in the prebiotic chemistry of early Earth^{39,40}. Analysis of the samples returned from the stardust mission revealed the presence of methylamine, ethylamine, along with glycine.⁴¹ Although methylamine (CH₃NH₂) has only been detected in the gas phase of the interstellar medium, laboratory experiments revealed that CH₃NH₂ could be formed via recombination of methyl (•CH₃) and amino (•NH₂) radicals in a matrix cage.⁴² Methane (CH₄) and ammonia (NH₃), which are precursors of •CH₃ and •NH₂ radicals, have been detected on icy grains;⁴³ hence methylamine (CH₃NH₂) is expected to exist on interstellar grains, albeit at a lower concentration.⁴⁴ Oxygen (O₂) was detected in the ISM

much later due to observational difficulties and probably low abundance in the gas phase of ISM.⁴⁵ It has been suggested that the poor abundance of oxygen in the gas phase could be due to the condensation of molecular oxygen (O₂) onto the interstellar grains.⁴⁶ Although the exact composition of molecular oxygen in the ice grains has not yet been determined, the possibility of O₂ being an active reactant in the ISM cannot be neglected. Very recently, Bergner et al. investigated the formation of methanol in methane (CH₄) and oxygen (O₂) ice mixture and reported that oxygen insertion chemistry should be efficient on grain surfaces in cold interstellar regions such as proto-stellar cores and protoplanetary disk midplanes.⁴⁷

Our laboratory experiments in astrophysical-like conditions revealed that aminomethanol could generate from amines on ice grains in proto-stellar cores or protoplanetary disks and subsequently incorporated inside comets or (and) meteorites,⁴⁸ which is (are) source(s) of extraterrestrial amino acids on early Earth. Aminomethanol (OHCH₂NH₂) could produce amino acid precursor – aminoacetonitrile (NH₂CH₂CN) in the presence of hydrogen cyanide; the later molecule could eventually produce amino acids on comets and meteorites through the Strecker route upon aqueous alteration. It is important to note here that the Strecker process could initiate even in the absence of liquid water. Quantum chemical calculations performed by Rimola et al. show that a Strecker-type reaction to form glycine is energetically feasible on solid-water ice grains.²³ The formation of acetonitrile precursor of glycine on an icy grain mantle via addition of hydrogen cyanide (HCN) to dehydrogenated form of aminomethanol has also been shown through calculations.⁴⁹ The present experimental study supports the theoretical investigations^{23,49} that have demonstrated the involvement of aminomethanol in Strecker synthesis of glycine in astrophysical environments.”

It is my feeling that the same experiment conducted using an ice mixture more representative of interstellar ices might not produce enormous quantities of aminomethanol, so the influence of the Strecker mechanism in feeding off of aminomethanol formed in precisely this way might not be so great (amino acetonitrile could be formed through other mechanisms, for example).

Response: The chemistry occurring in the interstellar medium is very complex. A given molecular entity could have multiple formation pathways in an actual or realistic interstellar ice mixture. Therefore, to understand the energetic feasibility and mechanisms of these processes, we need to first investigate simple model interstellar ice analogs before these studies can be extended to more complex systems.

Aminomethanol is a well-known intermediate of the Strecker process. Various theoretical studies have demonstrated the involvement of aminomethanol in Strecker synthesis of glycine in astrophysical-like conditions (Rimola, A., *et al. Phys. Chem. Chem. Phys.* 12, 5285-5294, (2010); Koch, D. M., *et al. J. Phys. Chem. C* 112, 2972-2980, (2008)). Our laboratory experiments suggest that aminomethanol could generate from amines on ice grains. The aminomethanol formed in proto-stellar cores or protoplanetary disks could be subsequently incorporated inside comets and meteorites, (Danger, G., *et al. Astrophys. J.* 756, 11, (2012)) which are sources of extraterrestrial amino acids on early earth. Aminomethanol (OHCH₂NH₂) could eventually produce amino acids on comets and meteorites via aminoacetonitrile following the Strecker route.

Relevant changes in the manuscript (on page 8): We have added the following sentence

“The present experimental study supports the theoretical investigations^{23,49} that have demonstrated the involvement of aminomethanol in Strecker synthesis of glycine in astrophysical environments.”

The Strecker mechanism should also arguably take place in the liquid phase, at least during the hydrolysis stage, so interstellar production in this way may not be so plausible. This should be commented on. Cometary, meteoritic or early Earth conditions may nevertheless lend themselves to amino acid production in this way.

Response: Strecker synthesis could also occur in solid-phase. Several theoretical investigations have been performed, which suggest that the Strecker process could happen in the absence of liquid water. For instance, quantum chemical calculations performed by Rimola *et al.* show that a Strecker-type reaction to form glycine is energetically feasible on solid-water ice grains (Rimola, A., *et al. Phys. Chem. Chem. Phys.* 12, 5285-5294, (2010)). The formation of acetonitrile precursor of glycine on an icy grain mantle via addition of hydrogen cyanide (HCN) to dehydrogenated form of aminomethanol has also been shown through calculations (Danger, G., *et al. Astrophys. J.* 756, 11, (2012)).

Relevant changes in the manuscript (on page 8): We have added the following sentences

*“It is important to note here that the Strecker process could initiate even in the absence of liquid water. Quantum chemical calculations performed by Rimola *et al.* show that a Strecker-type reaction to form glycine is energetically feasible on solid-water ice grains.²³ The formation of*

acetonitrile precursor of glycine on an icy grain mantle via addition of hydrogen cyanide (HCN) to dehydrogenated form of aminomethanol has also been shown through calculations.⁴⁹

Besides the above, I list a selection of comments, questions and suggestions:

1/ Considering who may be reading this work, I suggest that the title indicate clearly that the identification of aminomethanol has taken place in an experimental setting (rather than in space).

Response: We have modified the title to “Experimental Identification of Aminomethanol (NH₂CH₂OH) – The Key Intermediate in the Strecker Synthesis”.

Relevant changes in the manuscript (on page 1): *The title of the article is changed to “Experimental Identification of Aminomethanol (NH₂CH₂OH) – The Key Intermediate in the Strecker Synthesis”.*

2/ The relevance of electron bombardment in the interstellar context is not addressed. It should be briefly explained how these electrons are understood to come about and what is known about their energy spectrum.

Response: Astrophysical ices experience high energy radiations i.e. UV photons and cosmic-charged particles. The galactic cosmic ray field consists predominantly of protons with an energy distribution maximum of about 10 MeV and losses 99.9 % of their kinetic energy to the electron system of the target molecules. This electronic energy transfer generates energetic electrons (δ -electrons), which could penetrate into the ice up to a few hundred nanometers and initiate bond rupture processes in the molecular species of the ice mixture via inelastic energy transfer. In this experiment, we have used 5 keV electrons to mimic the effects of δ -electrons. The linear electron energy transfer of MeV protons to the ice target holds a similar value as the 5 keV electrons. Thus, our experiments simulate the formation of aminomethanol in the ice mixture of methylamine and oxygen via charged particles through electronic energy transfer.

Relevant changes in the manuscript (on pages 9-10): *We have added the following paragraph*

“Astrophysical ices experience high energy radiations i.e. UV photons and cosmic-charged particles. The galactic cosmic ray field consists predominantly of protons with an energy distribution maximum of about 10 MeV and losses 99.9 % of their kinetic energy to the electron system of the target molecules. This electronic energy transfer generates energetic electrons (δ -

electrons), which could penetrate into the ice up to a few hundred nanometers and initiate bond rupture processes in the molecular species of the ice mixture via inelastic energy transfer. In this experiment, we have used 5 keV electrons to mimic the effects of δ -electrons. The linear electron energy transfer of MeV protons to the ice target holds a similar value as the 5 keV electrons. Thus, our experiments simulate the formation of aminomethanol in the ice mixture of methylamine and oxygen via charged particles through electronic energy transfer.”

3/ What is the deposition angle of the ice? This is presumably not background deposition. The deposition angle can affect ice porosity and it is therefore relevant to state it (in exp. methods).

Response: The ice deposition angle is around 20° with respect to the normal of the substrate.

Relevant changes in the manuscript (on page 9): *We have added the following sentences*

“To prepare each ice mixture, gases of methylamine (CH_3NH_2 ; 99.95% Matheson TriGas) and oxygen molecule (O_2 ; 99.99% Sigma Aldrich) were premixed and deposited via a glass capillary array onto a silver substrate at an angle of 20° with respect to the normal of the substrate.”

4/ Figures 2 and S3 are not self-explanatory. More information should be provided in the caption to enable the reader to understand what is being shown. In Fig. 2, for instance, the plotted colors extend beyond the bounds of the plot, indicating some sort of three dimensional feature. It is not stated what this dimension and the coloring actually mean.

Response: We modified the figure captions of Figures 2 and S3 to provide a more detailed description of the figures. The graphs in Figure 2 and S3 are now replotted in 2D with grid lines to guide eyes and clearly show the observed masses.

Relevant changes in the manuscript: *Figures 2 and S3 are modified.*

Figure 2 caption *“PI-ReTOF mass spectra measured as a function of temperature during the during the temperature program desorption phase of irradiated methylamine (CH_3NH_2)-oxygen (O_2) ice mixtures at photon energies of (a) 9.50 and (b) 9.10 eV. The dark purple colored lines indicate ion counts.”*

Figure S3 caption “PI-ReTOF mass spectrum measured as a function of temperature during the TPD phase of non-irradiated methylamine (CH_3NH_2) and oxygen (O_2) ice mixture at a photon energy of 10.49 eV. The dark colored purple lines indicate ion counts.”

5/ While it is experimentally interesting and laudable to have formed and identified aminomethanol, the meaning in an interstellar context is less clear, and more should be made of this. For example, interstellar ices should have substantial water content, while alternative C-H bonds for insertion by O singlet-D should also be prevalent in methane and methanol, and arguably others. We also do not know what the O_2 composition of interstellar ices actually should be, based on observations (although this is not the case for cometary ice). It could, in fact, be small; smaller even than that of NH_2CH_3 , which has also not yet been detected in the solid phase in the ISM and has only recently been detected in the gas phase in rather small quantities. How might all of this impact the importance of the findings?

Response: Please confer to our reply to referee 1. Amines like methylamine and ethylamine have been observed in the interstellar medium towards SgrB2 and Ori A (Fourikis, N., *Astrophys. J.* 191, L139, (1974); Kaifu, N., *Astrophys. J.* 198, L85, (1975)). Both the amines were also identified on comet 67 p/Churyumov-Gerasimenko (Goesmann, F. et al., *Science* 349, aab0689, (2015)). Analysis of the samples returned from the *stardust* mission revealed the presence of methylamine, ethylamine, along with glycine (Glavin, D. P., et al. *Meteorit. Planet Sci.* 43, 399-413, (2008)). These astronomical observations combined with theoretical and experimental studies suggest that simple amines like methylamine can be a precursor of amino acids. We agree with the reviewer that methylamine has only been detected in the gas phase of the interstellar medium. However, recent laboratory experiments showed explicitly that methylamine could be formed via recombination of neighboring methyl and amino radicals in a matrix cage (Förstel, M., et al. *Astrophys. J.* 845, 83, (2017)). Methane and ammonia, which are precursors of methyl and amino radicals, have been detected on icy grains (Gürtler, J. et al. *A&A* **390**, 1075-1087 (2002)); hence methylamine is expected to exist on interstellar grains, albeit at a lower concentration.

Oxygen (O_2) was detected in the ISM (Larsson, B. et al., *A&A* **466**, 999-1003 (2007)) much later due to observational difficulties and probably low abundance in the gas phase of ISM. It has been suggested that the poor abundance of oxygen in the gas phase could be due to the condensation of molecular oxygen (O_2) onto the interstellar grains (Du, F., et al. *A&A* 538, A91 (2012)).

Although the exact composition of molecular oxygen in the ice grains has not yet been determined, the possibility of O₂ being an active reactant in the ISM cannot be neglected. Very recently, Bergner et al. investigated the formation of methanol in methane (CH₄) and oxygen (O₂) ice mixture and reported that oxygen insertion chemistry should be efficient on grain surfaces in cold interstellar regions such as proto-stellar cores and protoplanetary disk midplanes (Bergner, J. B., *et al.* *Astrophys. J.* 845, 29, (2017)). Formation of COMs by the oxygen insertion method does not require diffusion of heavy molecular species indicating that the oxygen insertion reactions will be more facile at temperatures lower than 30 K. We agree with the reviewer that alternative reactions of oxygen such as insertion of O (¹D) atom at the C-H bonds of methane and methanol should be prevalent in the ISM. However, that does not imply that the probability of oxygen reacting with amines to form aminomethanol (OHCH₂NH₂) will be negligible.

Furthermore, aminomethanol undergoes unimolecular decomposition to form methanimine (CH₂NH) in the presence of water. In a water-rich ice, the surrounding water molecules could act as a catalyst in the dehydration process of aminomethanol (Rimola, A., *et al.* *Phys. Chem. Chem. Phys.* 12, 5285-5294, (2010)); this could also be the reason for the failure in detecting aminomethanol in a water-rich interstellar environment. Here, we choose a non-aqueous ice mixture to avoid the dissociation process of aminomethanol in the ice matrix.

Relevant changes in the manuscript (on pages 7-8): We have added the following paragraphs to discuss the significance of our results in the context of the interstellar medium.

“Primary amines such as methylamine (CH₃NH₂) and ethylamine (C₂H₅NH₂) were observed in ISM towards SgrB2 as well as on comet 67 p/Churyumov-Gerasimenko.³⁶⁻³⁸ Amines are also contemplated to play a critical role in the prebiotic chemistry of early Earth^{39,40}. Analysis of the samples returned from the stardust mission revealed the presence of methylamine, ethylamine, along with glycine.⁴¹ Although methylamine (CH₃NH₂) has only been detected in the gas phase of the interstellar medium, laboratory experiments revealed that CH₃NH₂ could be formed via recombination of methyl (•CH₃) and amino (•NH₂) radicals in a matrix cage.⁴² Methane (CH₄) and ammonia (NH₃), which are precursors of •CH₃ and •NH₂ radicals, have been detected on icy grains;⁴³ hence methylamine (CH₃NH₂) is expected to exist on interstellar grains, albeit at a lower concentration.⁴⁴ Oxygen (O₂) was detected in the ISM much later due to observational

difficulties and probably low abundance in the gas phase of ISM.⁴⁵ It has been suggested that the poor abundance of oxygen in the gas phase could be due to the condensation of molecular oxygen (O₂) onto the interstellar grains.⁴⁶ Although the exact composition of molecular oxygen in the ice grains has not yet been determined, the possibility of O₂ being an active reactant in the ISM cannot be neglected. Very recently, Bergner et al. investigated the formation of methanol in methane (CH₄) and oxygen (O₂) ice mixture and reported that oxygen insertion chemistry should be efficient on grain surfaces in cold interstellar regions such as proto-stellar cores and protoplanetary disk midplanes.⁴⁷

Our laboratory experiments in astrophysical-like conditions revealed that aminomethanol could generate from amines on ice grains in proto-stellar cores or protoplanetary disks and subsequently incorporated inside comets or (and) meteorites,⁴⁸ which is (are) source(s) of extraterrestrial amino acids on early Earth. Aminomethanol (OHCH₂NH₂) could produce amino acid precursor – aminoacetonitrile (NH₂CH₂CN) in the presence of hydrogen cyanide; the later molecule could eventually produce amino acids on comets and meteorites through the Strecker route upon aqueous alteration.”

Relevant changes in the manuscript (on page 3): *We have added the following sentences “Furthermore, the rate of unimolecular decomposition of aminomethanol (OHCH₂NH₂) to methanimine (CH₂NH) will significantly reduce in a non-aqueous medium containing reactants methylamine and oxygen. In a water-rich environment, the surrounding water molecules could act as a catalyst in the dehydration process of aminomethanol²³ and therefore could obscure its detection.”*

6/ How applicable is Strecker synthesis in a purely solid-phase context?

Response: As mentioned above, Strecker synthesis could occur in solid-phase. Several theoretical investigations have been performed, which suggest that Strecker’s process could happen in the absence of liquid water.

Relevant changes in the manuscript (on page 8): *We have added the following sentences*

“It is important to note here that the Strecker process could initiate even in the absence of liquid water. Quantum chemical calculations performed by Rimola et al. show that a Strecker-type reaction to form glycine is energetically feasible on solid-water ice grains. Formation of

acetonitrile and aminomethanol precursors of glycine on an icy grain mantle has also been demonstrated through calculations.”

7/ How does the mechanism tested here compare with other lab studies, e.g. that of Elsila et al. (2007, ApJ, 600, 911), who used H₂O:CH₃OH:HCN:NH₃ mixtures to produce amino acids? Although they considered a slightly different mechanism, their isotopic labeling results indicated that Strecker synthesis should be less important than radical reaction processes.

Response: The purpose of this study is to (1) investigate the formation of the aminomethanol – a crucial intermediate of Strecker's process in well-defined and controlled laboratory simulation experiments, (2) produce the target molecule in sufficient abundance to enable its detection in the gas phase. As mentioned before, aminomethanol is very labile and easily decomposes to imine in the presence of water. Neighboring water molecules act as a catalyst in the dehydration process of aminomethanol and therefore could obscure its detection. Unlike previous laboratory studies, here we choose a non-aqueous ice mixture to avoid the unimolecular decomposition of aminomethanol in the ice matrix. In our experiments, the aminomethanol (OHCH₂NH₂) could either form via O(¹D) atom insertion into the C-H bond of methylamine or radical-radical recombination of hydroxy and •CH₂NH₂ radicals. The former pathway could be dominant as it is energetically more favorable.

8/ The identification of aminomethanol rests strongly on the calculations, which mainly take a back seat in the text. Are there any alternative estimates or measurements of any of the ionization energies calculated here? Is there any alternative verification that can now or in future be used to back up the calculations?

Response: Previous work (Turner et al. ChemPhysChem, 2021, 22, 985-994) at this level of theory (CCSD(T)/CBS+ZPE) has been able to match experiment to within 0.05 eV in all known or even 0.01 eV in many cases. Hence, the IEs are similarly accurate here.

If all of the above points are addressed, I feel that the manuscript would be ready for publication.

Reviewer #3 (Remarks to the Author):

The present communication entitled “Gas-phase identification of aminomethanol (NH₂CH₂OH) – the key intermediate in the Strecker synthesis”, S.K. Singh et al shows an experimental work on the formation of NH₂CH₂OH induced by keV electrons of NH₂CH₃:O₂ ice. Coupling VUV photoionization time of flight spectrometer and theoretical calculations (IE of different isomers), they have identified NH₂CH₂OH isomers desorbing from irradiated ice. This work relies on well established procedure in Kaiser’s laboratory to investigate formation of different molecules in irradiated ices.

A weak aspect of the paper is that the investigation of NH₂CH₂OH formation is not a novelty from a conceptual point of view. In the literature, one can find several studies dedicated to investigate the formation of aminomethanol experimentally and theoretically (several references are in the present article). For example it has been shown that NH₂CH₂OH can be formed in H₂O:NH₃:H₂CO ice (F Duvernay et ApJ 791 (2014) 75).

Response: In the following article (F Duvernay et ApJ 791 (2014) 75), the authors have measured the IR spectrum of the organic residue that remains after sublimation of the reactants and compared with the calculated IR spectrum of aminomethanol and its clusters (CH₂NH₂OH)₁₀. It is important to note here that IR spectroscopy alone cannot distinguish the complex organic molecules in a multicomponent ice mixture due to overlapping infrared absorptions of the products and reactants; IR spectroscopy can only identify functional groups in complex mixtures, but not individual organic molecules (Turner, A. M. & Kaiser, R. I. *Acc. Chem. Res.* 53, 2791-2805, (2020); Singh, S. K. & Kaiser, R. I. *Chem. Phys. Lett.* 766, 138343, (2021)).

Furthermore, the mass spectra measured by the authors through electron impact ionization technique at 70 eV caused extensive fragmentation of the parent ion. In addition, the electron impact ionization mass spectrometry cannot be used to distinguish the structural isomers and determine the true structure of the products. Therefore, the assignment of aminomethanol in these articles is not affirmative. This has been mentioned in the introduction section of the manuscript on page 3.

Here, we have employed isomer-selective photoionization technique to detect the aminomethanol in methylamine and oxygen ice mixture. This technique has been very effective in determining the nature of the complex organic molecules in multicomponent ice mixtures.

It would be worth mention why in particular, $\text{NH}_2\text{CH}_3\text{:O}_2$ is particularly interesting to investigate aminomethanol formation.

Response: Aminomethanol easily decomposes to methanimine (CH_2NH) in an aqueous environment. In a water-rich environment, the surrounding water molecules could act as a catalyst in the dehydration process of aminomethanol (Rimola, A., *et al. Phys. Chem. Chem. Phys.* 12, 5285-5294, (2010)) and could obscure its detection. Therefore, unlike previous laboratory studies, here we choose a non-aqueous ice mixture to avoid the unimolecular decomposition of aminomethanol in the ice matrix.

Furthermore, the oxygen-insertion chemistry in interstellar-like conditions have not been extensively explored. This novel pathway is energetically favorable to directly convert simple organic molecules to complex organic molecules. Unlike radical-radical recombination pathway, oxygen insertion does not require diffusion of heavy molecular species and therefore should be more active in colder regions of interstellar medium such as proto-stellar cores. Very recently, Bergner *et al.* (*Astrophys. J.* 845, 29, 2017) have investigated the formation of methanol in interstellar-like condition via oxygen insertion at the C-H bond of methane. In order to understand the energetic feasibility, efficiency, and complexity of oxygen-insertion pathways in ISM, it is important to investigate the chemical processes in processed astrophysical-ice analogs containing oxygen as one of the components.

Relevant changes in the manuscript (on page 3): *We have added the following sentences “Furthermore, the rate of unimolecular decomposition of aminomethanol (OHCH_2NH_2) to methanimine (CH_2NH) will significantly reduce in a non-aqueous medium containing reactants methylamine and oxygen. In a water-rich environment, the surrounding water molecules could act as a catalyst in the dehydration process of aminomethanol²³ and therefore could obscure its detection.”*

Another point, the whole procedure of isomer identification relies on theoretical calculations. Authors considered error limits due theoretical uncertainties and experimental conditions for benchmark systems. It would be worth also to compare with experimental IE values of discussed isomers (if available).

Response: While we agree with the reviewer that having such data on hand for these molecules would be beneficial, since they are not available, we must rely on benchmarks; this is a well-

established approach in computational and experimental physical chemistry, when experimental data are absent. As given in our response to Reviewer 2, Comment 8, previous benchmarks show that this computational approach has exceptional experimental fidelity. The molecules explored herein provide no questionable results or behaviors leading us to question the extension of these methods to the present systems.

One can find also inconsistency between the text in “Mass spectrometry” paragraph and Fig 2. It has been stated the data recorded at 9.50 eV reveal several products at m/z 45, 47 58 and 60. This is visible in Fig. 2b (obtained for 9.10 eV if we follow a caption)? Figure 2 is nice to follow globally dynamics of appearing and disappearing m/z species as a function of temperature. However, the present figure does not allow to read masses in details. Lines to guide eyes should be added or it would be better to show a cut for 210 K –intensity as a function of m/z for all three ices $\text{NH}_2\text{CH}_3:\text{O}_2$, $\text{Nd}_2\text{CD}_3:\text{O}_2$ and $\text{NH}_2\text{CH}_3:18\text{O}_2$. A short comment of high mass formation would enrich the article.

Response: Ion counts at m/z ratios of 45, 47, 58, and 60 are also observed in the spectrum measured at 9.50 eV. The graphs in Figure 2 are now replotted in 2D with grid lines to clearly show the observed masses.

Higher molecular weights products are observed at $m/z = 58, 59, 60, 61, 74,$ and 90 . Signals at $m/z = 60$ ($\text{C}_2\text{H}_8\text{N}_2$), 59 ($\text{C}_3\text{H}_9\text{N}$) and 58 ($\text{C}_2\text{H}_6\text{N}_2$) were previously observed in processed pure methylamine ice and can be tentatively assigned to isomers of ethylene diamine, N-methyl-ethanamine and N-methylformimidamide (. A possible formation pathway of ethylene diamine could follow a radical-radical recombination of two CH_2NH_2 radicals. N-methyl-ethanamine could generate via recombination of $\bullet\text{CH}_3$ and $\bullet\text{CH}_2\text{NHCH}_3$ radicals. While N-methylformimidamide could generate via dehydrogenation of ethylene diamine ($\text{C}_2\text{H}_8\text{N}_2$). Oxygen could insert at the C-H and/or N-H bonds of N-methylformimidamide to generate products observed at $m/z = 74$ ($\text{C}_2\text{H}_6\text{N}_2\text{O}$) and 90 ($\text{C}_2\text{H}_6\text{N}_2\text{O}_2$). It important to note here that mass signals 47, 61, 74 and 90 were not observed in pure methylamine ices.

Relevant changes in the manuscript (on page 5): Data recorded at 9.50 eV reveal prominent product signal at mass-to-charge ratios (m/z) of 42 (CH_2N_2), 43 (C_2NH_5^+), 45 ($\text{CH}_3\text{NO}/\text{C}_2\text{H}_7\text{N}$), 47 (CH_5NO), 58 ($\text{C}_2\text{H}_6\text{N}_2$), 59 ($\text{C}_3\text{H}_9\text{N}$), 60 ($\text{C}_2\text{H}_8\text{N}_2$), 61 (CH_3NO_2), 74 ($\text{C}_2\text{H}_6\text{N}_2\text{O}$), and 90 ($\text{C}_2\text{H}_6\text{N}_2\text{O}_2 / \text{C}_3\text{N}_3\text{H}_{12}$) along with the reactant signal at $m/z = 31$ (CH_3NH_2). Signals at $m/z = 60$ ($\text{C}_2\text{H}_8\text{N}_2$), 59 ($\text{C}_3\text{H}_9\text{N}$) and 58 ($\text{C}_2\text{H}_6\text{N}_2$) were previously observed in processed pure methylamine ice and can be tentatively assigned to isomers of ethylene diamine, N-methyl-

ethanamine and N-methylformimidamide.³³ A possible formation pathway of ethylene diamine ($C_2H_8N_2$) could follow a radical-radical recombination of two $\bullet CH_2NH_2$ radicals. N-methyl-ethanamine could generate via recombination of $\bullet CH_3$ and $\bullet CH_2NHCH_3$ radicals. While N-methylformimidamide ($C_2H_6N_2$) could generate via dehydrogenation of ethylene diamine ($C_2H_8N_2$). Oxygen could insert at the C-H and/or N-H bonds of N-methylformimidamide ($C_2H_6N_2$) to generate products observed at $m/z = 74$ ($C_2H_6N_2O$) and 90 ($C_2H_6N_2O_2$). It important to note here that mass signals 47, 61, 74 and 90 were not observed in pure methylamine ices.³³

It would be also worth to clarify broadening of the TPD profile at $m/z = 49$. It has been mentioned that is due fragmentation of masses $m/z = 58$ ($C_2H_6N_2$) and 80 ($CH_4N_2O_2$) (e.g. loss of fragments of $m/z = 9$ and 31 ?, respectively). This fragmentation is caused by which factor? Moreover, why we do not observe similar broadening for other ices?

Response: Higher molecular weight products could fragment during the photoionization process if the ionic state is shallow and photons have sufficient energy to cross the dissociation limit; the ionic fragments will appear in the lower mass channels. Broadening of the TPD profile at $m/z = 49$ in the $CH_3NH_2 - ^{18}O_2$ ice close to 200 and 250 K could be due to the fragmentation of higher masses observed at $m/z = 91$ ($C_2H_3N_2^{18}O_2$) and 80 ($CH_4N_2^{18}O_2$). Product observed at mass 91 could fragment to $CH_2N_2^+$ ($m/z = 42$) and HCO_2^+ ($m/z = 49$) ions, the later fragment could add to the ion counts at $m/z = 49$ around 200 K temperature. Similarly, product observed at $m/z = 80$ could fragment to generate $N_2H_3^+$ ($m/z = 31$) and HCO_2^+ ($m/z = 49$) ions; the later fragment will increase the ion counts at $m/z = 49$ near 250 K causing broadening of the TPD profile of our target molecule. In $CH_3NH_2 - O_2$ and $CD_3ND_2 - O_2$ ice systems, the ion fragment HCO_2^+ will appear at m/z ratio of 45 and 46 respectively and therefore cannot appear in the mass channel of our target molecule.

Relevant changes in the manuscript (on page 6): We have added the following sentences “The broadening of the TPD profile at $m/z = 49$ in the $CH_3NH_2 - ^{18}O_2$ ice close to 200 and 250 K could be due to the fragmentation of higher masses observed at $m/z = 91$ ($C_2H_3N_2^{18}O_2$) and 80 ($CH_4N_2^{18}O_2$) (Figure S4). Product observed at mass 91 could fragment to $CH_2N_2^+$ ($m/z = 42$) and HCO_2^+ ($m/z = 49$) ions, the later fragment could add to the ion counts at $m/z = 49$ around 200 K. Similarly, product observed at $m/z = 80$ could fragment to generate $N_2H_3^+$ ($m/z = 31$) and HCO_2^+ ($m/z = 49$) ions; the later fragment will increase the ion counts at $m/z = 49$ near 250 K

causing broadening of the TPD profile of our target molecule. In $\text{CH}_3\text{NH}_2 - \text{O}_2$ and $\text{CD}_3\text{ND}_2 - \text{O}_2$ ice systems, the ion fragment HCO_2^+ will appear at m/z ratio of 45 and 46 respectively and therefore cannot appear in the mass channel of our target molecule.”

Relevant changes in the supporting information (on page 7): Figure S4 is modified.

REVIEWERS' COMMENTS

Reviewer #2 (Remarks to the Author):

I thank the authors for their consideration of my comments and for making adjustments to the manuscript.

I have two final points:

1) In the new text:

"The galactic cosmic ray field consists predominantly of protons with an energy distribution maximum of about 10 MeV and losses 99.9 % of their kinetic energy to the electron system of the target molecules."

...a reference should be provided for these numbers.

2) The authors responded to my point #8 with the following reply:

"Response: Previous work (Turner et al. ChemPhysChem, 2021, 22, 985-994) at this level of theory (CCSD(T)/CBS+ZPE) has been able to match experiment to within 0.05 eV in all known or even 0.01 eV in many cases. Hence, the IEs are similarly accurate here."

A sentence stating something along these lines should be added somewhere in the text.

Reviewer #3 (Remarks to the Author):

In revised version of the manuscript, the authors have addressed all of the points I raised in my review. I am satisfied with their answers and the present form of the manuscript. Therefore, I recommend the article to be published.

Reviewer #2 (Remarks to the Author):

I thank the authors for their consideration of my comments and for making adjustments to the manuscript.

I have two final points:

1) In the new text:

"The galactic cosmic ray field consists predominantly of protons with an energy distribution maximum of about 10 MeV and losses 99.9 % of their kinetic energy to the electron system of the target molecules."

...a reference should be provided for these numbers.

Response: We have added the following reference "Bennett, C. J., Jamieson, C. S., Osamura, Y. & Kaiser, R. I. A Combined experimental and computational investigation on the synthesis of acetaldehyde [CH₃CHO(X¹A')] in interstellar ices. *Astrophys. J.* 624, 1097-1115, (2005)."

2) The authors responded to my point #8 with the following reply:

"Response: Previous work (Turner et al. *ChemPhysChem*, 2021, 22, 985-994) at this level of theory (CCSD(T)/CBS+ZPE) has been able to match experiment to within 0.05 eV in all known or even 0.01 eV in many cases. Hence, the IEs are similarly accurate here."

A sentence stating something along these lines should be added somewhere in the text.

Response: Following sentence has been added in the main manuscript on page 5 "*It is important to stress here that the calculated IEs of the isomers can be lower by 0.05 eV or higher by 0.02 eV based on a comparison of experimental and calculated IEs of molecular benchmark systems.³¹*" to support our response and previous work (Turner et al. *ChemPhysChem*, 2021, 22, 985-994) has been cited.

Reviewer #3 (Remarks to the Author):

In revised version of the manuscript, the authors have addressed all of the points I raised in my review. I am satisfied with their answers and the present form of the manuscript. Therefore, I recommend the article to be published.

Response: Thank you.